# Preclinical Application of Reduced Manipulated Processing Strategy to Collect Transplantable Hepatocytes: A Pilot and Feasibility Study

**DOI:** 10.3390/jpm11050326

**Published:** 2021-04-21

**Authors:** Ya-Hui Chen, Hui-Ling Chen, Cheng-Maw Ho, Hung-Yen Chen, Shu-Li Ho, Rey-Heng Hu, Po-Huang Lee, Mei-Hwei Chang

**Affiliations:** 1Hepatitis Research Center, National Taiwan University Hospital, Taipei 100, Taiwan; tiiy2000@yahoo.com.tw (Y.-H.C.); hlchen9@ntu.edu.tw (H.-L.C.); p311308@hotmail.com (H.-Y.C.); 2Department of Pediatrics, National Taiwan University Children Hospital, Taipei 100, Taiwan; changmh@ntu.edu.tw; 3Department of Surgery, National Taiwan University Hospital and National Taiwan University College of Medicine, Taipei 100, Taiwan; rhhu@ntu.edu.tw (R.-H.H.); pohuang1115@ntu.edu.tw (P.-H.L.); 4Department of Anatomy and Cell Biology, National Yang Ming Chiao Tung University, Taipei 112, Taiwan; lilybox.529@hotmail.com

**Keywords:** hepatocyte, reduced manipulated strategy, cell processing, cell therapy

## Abstract

Background: The complex isolation and purification process of hepatocytes for transplantation is labor intensive and with great contamination risk. Here, as a pilot and feasibility study, we examined in vitro and in vivo hepatocyte isolation feasibility and cell function of Cell Saver^®^ Elite^®^, an intraoperative blood-cell-recovery system. Methods: Rat and pig liver cells were collected using this system and then cultured in vitro, and their hepatocyte-specific enzymes were characterized. We then transplanted the hepatocytes in an established acute liver–injured (retrorsine+D-galactosamine-treated) rat model for engraftment. Recipient rats were sacrificed 1, 2, and 4 weeks after transplantation, followed by donor-cell identification and histological, serologic, and immunohistopathological examination. To demonstrate this Cell Saver^®^ strategy is workable in the first place, traditional (classical) strategy, in our study, behaved as certainty during the cell manufacturing process for monitoring quality assurance throughout the course, from the start of cell isolation to post-transplantation. Results: We noted that in situ collagenase perfusion was followed by filtration, centrifugation, and collection in the Cell Saver^®^ until the process ended. Most (>85%) isolated cells were hepatocytes (>80% viability) freshly demonstrating hepatocyte nuclear factor 4α and carbamoyl-phosphate synthase 1 (a key enzyme in the urea cycle), and proliferating through intercellular contact in culture, with expression of albumin and CYP3A4. After hepatocyte transplantation in dipeptidyl peptidase IV (−/−) rat liver, wild-type donor hepatocytes engrafted and repopulated progressively in 4 weeks with liver functional improvement. Proliferating donor hepatocyte–native biliary ductular cell interaction was identified. Post-transplantation global liver functional recovery after Cell Saver and traditional methods was comparable. Conclusions: Cell Saver^®^ requires reduced manual manipulation for isolating transplantable hepatocytes.

## 1. Introduction

In patients with acute liver failure, hepatocyte transplantation is a promising, safe and less invasive procedure alternative to liver transplantation [1] Animal studies have clearly proven the efficacy of hepatocyte transplantation; however, this has not translated into clinical practice where the benefit is often limited [2,3]. One of the difficulties that limit wide-spread application of hepatocyte transplantation is the complex isolation and purification process that requires a clinical-grade good manufacturing practice (GMP)-approved laboratory [4]. Moreover, the repetitive transfer of the “intermediate cell product” in and out of the centrifuge during cell processing is labor intensive and greatly increases contamination risk. For reducing the processing time and thus the degree of cell ischemia, a semiautomatic closed-circuit system would be exceedingly helpful.

Cell Saver^®^ Elite^®^ (Haemonetics, Braintree, MA, USA) is an intraoperative cell recovery machine originally used for autologous red blood cell transfusion [5]. This system uses a bowl centrifuge technology [6], which is different from the general centrifuge and has the advantage of processing a large amount of cells in a short period. We adopted this strategy for rapid, nondestructive separation of liver cells from a nonviable matrix by using reduced manipulated cells to demonstrate the potential of this approach in next-generation clinical cell therapies. The term of reduced manipulated strategy here is referred to an effort to reduce manual laboring work in hood or to reduce steps that required heavy GMP laboratory support, although in most jurisdictions, minimal manipulation is defined as acts that do not result in fundamental changes to the structure or biological characteristics of the original cells or tissues [7].

We hypothesized that liver cell isolation using the Cell Saver^®^ strategy is feasible, and the isolated cells are functional and transplantable. Therefore, as a pilot and feasibility study, we aimed to, in vitro, characterize liver cells isolated using this strategy with a scale-up attempt to large animals, and in vivo, transplant these cells into acute liver injured rats [8]. To demonstrate this Cell Saver^®^ strategy is workable in the first place, traditional (classical) strategy, in our study, behaved as certainty during the cell manufacturing process for monitoring quality assurance throughout the course, from the start of cell isolation to post-transplantation.

## 2. Methods

### 2.1. Animals and Ethics Statement

Male F344 rats aged 8 weeks and weighing 200–250 g purchased from the National Laboratory Center (Taipei, Taiwan) were used as donor animals. Male dipeptidyl peptidase IV-deficient [DPPIV (−/−)] rats, kindly provided by Professor Sanjeev Gupta from the Albert Einstein College of Medicine, were used as recipient animals. With an attempt to scale up cell isolation to large animals, miniature pigs (weight = 20 kg) were obtained from the Experimental Farm of National Taiwan University. Rats and pigs were in-house bred and maintained on standard laboratory chow and daily 12-h light/dark cycles. Donor rats were anesthetized with ketamine/xylazine (75/5 mg/kg) through intramuscular injection before in situ collagenase perfusion. Pigs were sedated with Zoletil (Virbac, Carros, France) (2–4 mg/kg) and xylazine (GWF Nutrition, Wiltshire, UK) (2 mg/kg) and then administered propofol (Fresenius Kabi Austria GmbH, Graz, Austria) or isoflurane (Baxter, San Juan, Puerto Rico). The study was approved by the Institutional Animal Care and Use Committee (Approval No: 20160136 and 20180171) of the College of Medicine and College of Public Health, National Taiwan University. All animals received humane care in accordance with the guidelines of the National Science Council of Taiwan (1997) as well as the criteria outlined in the Guide for the Care and Use of Laboratory Animals (National Institutes of Health publication 86–23, 1985 revision). Moreover, all procedures were performed in accordance with these guidelines.

### 2.2. Experimental Design

Figure 1 illustrates the scheme of the study design. In this study, the latter half of the cell processing procedure was technically modified, namely purification and cell collection. The Cell Saver^®^ strategy replaced tedious and time-consuming traditional steps (filtration, repeated cell washing with centrifugation, and final cell collection) that required multiple transfers of intermediate cell products with shuttling between the hood and room, posing a high microbial contamination risk.

### 2.3. Hepatocyte Isolation and Phenotypic Characterization

Rat hepatocytes were isolated through in situ liver perfusion and collagenase digestion as previously described [8]. The gross digested liver tissue was removed in bloc and then processed using the traditional method [8] or Cell Saver^®^ strategy. Pig hepatocytes were isolated through ex vivo perfusion and collagenase digestion of partial liver lobe after en bloc liver removal (Appendix A) followed by processing through the Cell Saver^®^ strategy. The Cell Saver^®^ strategy (which includes filtration, 3-min washing with centrifugation, and cell collection) was applied to the latter half of the cell processing procedure. The traditional technique involves manual filtration by using a siege of 100 μm followed by repeated cycles of differential centrifugation (50 g) and pellet resuspension for at least three times and Percoll gradient purification. [8]

The viability and purity of each isolation round were assessed using trypan blue (Sigma, St. Louis, MO, USA) exclusion on a Bright-Line hemocytometer (Sigma-Aldrich, St. Louis, MO, USA). Immunocytological analysis immediately after isolation was used for phenotypic characterization of released cellular products. Mature hepatocytes were marked with nuclear expression of hepatocyte nuclear factor 4 alpha (HNF4α) and with functional expression of a key enzyme in the urea cycle carbamoyl phosphate synthetase 1 (CPS-1). Macrophages were marked with CD163 expression. Cholangiocytes and sinusoidal endothelial cells were marked with CK-19 and SE-1, respectively.

### 2.4. Cell Culture

Isolated hepatocytes, if not transplanted, were inoculated in a 24-well culture plate coated with type I collagen at a density of 1.75 × 10^5^ cells/well for examining cell characteristics and proliferation. Liver cell culture was performed according to a previously described protocol [9]. Water soluble tetrazolium-1 (WST-1) cell proliferation assay (Sigma-Aldrich) was performed according to the manufacturer’s protocol.

### 2.5. Hepatocyte Transplantation

Acute liver injured rat models [retrorsine (R) + D-galactosamine (D)] were used for hepatocyte transplantation. The working solutions of retrorsine (Sigma) and D-galactosamine (Sigma) were prepared as described previously [8] and used immediately. The animal model of hepatocyte transplantation (i.e., R+D) was described previously [8], and an experimental design for hepatocyte transplantation is shown in Appendix A. In brief, male DPPIV (−/−) rats received two treatments of retrorsine (30 mg/kg, intraperitoneal injection [IP]) 2 weeks apart at 6 and 8 weeks of age. Acute hepatic injury was induced with D-galactosamine treatment (0.7 g/kg, IP) 2 weeks after the second retrorsine treatment.

Isolated rat hepatocytes (1 × 10^7^/mL) from wild-type F344 rats prepared using Cell Saver^®^ and traditional strategies were transplanted intraportally through the cap of a 24-G catheter (diameter = 0.7 mm; Insyte, BD) in 70 s at 24 h after D-galactosamine injection [10].

As a comparative group, male DPPIV (−/−) rats received hepatocytes isolated through the traditional method. A sham procedure was performed on rats, which involved L-15 solution infusion without cells. For the Cell Saver^®^ group, the surviving rats were sacrificed, and their livers were harvested at the time points of 1, 2, and 4 weeks after hepatocyte transplantation, whereas for the sham group, the time points were 1 and 2 weeks.

One piece of liver tissue from the right lobe was fixed in 4% formaldehyde and was paraffin-embedded for histology. The other pieces from each liver lobe were snap frozen in liquid nitrogen or were embedded into an optimum cutting temperature compound and stored at −80 °C.

### 2.6. Hepatic Histology and Determination of Liver Repopulation

Paraffin sections of liver tissue were stained with hematoxylin–eosin for evaluating histopathological changes. Transplanted hepatocytes in the recipient liver were identified using enzyme histochemical staining for DPPIV in liver cryosections as previously described [8,11].

To analyze liver repopulation, three to four sections from multiple liver lobes per rat were stained for DPPIV histochemical activity. Microphotographs were obtained from consecutively adjacent areas to include the whole section under 100× magnification by using a digital camera (BX53, Olympus, Tokyo, Japan) with SPOT Imaging Solutions (Diagnostic Instruments, Sterling Heights, MI, USA). The area occupied by the transplanted hepatocytes was quantitated with Image J (National Cancer Institute, Bethesda, MD, USA).

### 2.7. Histochemistry and Immunohistochemistry

All histochemical and immunohistochemical staining procedures were performed according to previously described protocols [11]. Ductular proliferative cells were marked with cytokeratin 19 (CK-19) expression [11]. Double immunofluorescence staining procedures were applied for detecting DPPIV and CK-19. Primary antibodies were previously described [11] and are summarized in Table 1. Appropriate secondary antibodies used in various experiments were Alexa Fluor 488 donkey anti-mouse IgG (Molecular Probes, Eugene, OR, USA) and Alexa Fluor 594 donkey anti-goat IgG (Molecular Probes). Nuclei were labeled with 4′,6-diamidino-2-phenylindole (Molecular Probes) [10,11].

### 2.8. Serological Assay

Hepatic venous blood was sampled after recipient rats were sacrificed. Biochemical analyses were performed in an animal laboratory by using standard automated assays as previously described [10].

### 2.9. Statistical Analysis

Data are shown in a qualitative manner or presented as the mean ± standard error of the mean. The significance of differences was analyzed through a t-test by using SPSS (version 13.0; SPSS, Chicago, IL, USA). *p* < 0.05 was considered statistically significant.

## 3. Results

### 3.1. Liver Cell Isolation with Cell Saver^®^ Strategy

Cell isolation process by using the Cell Saver^®^ strategy is shown in Figure 2. After in situ collagenase perfusion, liver was removed en bloc and minced to release liver cells. Phosphate buffered saline was added to lower the enzyme activity and stop the digestion. The gross connective tissue was removed en bloc, and the “liver soup” was aspirated using a suction tube directly into the Cell Saver^®^ closed-circuit system, which was first filtered (Figure 2A) through continuous washing of cells for 3 min with centrifugation (Figure 2B); ready-to-use cell suspension was collected into a bag (Figure 2C), which was separated from the waste solution (Figure 2D). The whole sterile closed-system processing from aspiration of liver soup to bag product, in our pilot study, took less than 30 min. Hepatocytes isolated from rats or pigs constituted the major cell population (>85%) in end cell products, and hepatocyte viability was >80%. Specifically, rat hepatocytes recovered from Cell Saver^®^-implemented procedure were 3.0 × 10^8^ per rat with 82.8% viability, and from traditional method, 3.2 × 10^8^ per rat with 85.0% viability. The number of isolation processes using Cell Saver^®^-implemented procedure was two rounds for pig donors (one donor one time) and three rounds for rat donors (three donors one round). Rat liver cell products were then transfused into the portal vein of recipient rats with acute liver injury in vivo or sent for in vitro assays.

### 3.2. Characterization of Isolated Cells through In Vitro Assessment

Phenotypic characterization of released cellular products immediately after isolation of rat livers is shown in Figure 3. Both groups demonstrated high concentrations of mature and functional hepatocytes (HNF4α^+^ and CPS-1^+^). Few CD163^+^ macrophages (Figure 3), cholangiocytes, and sinusoidal endothelial cells (Appendix A) could be observed in either group.

The primary hepatocyte culture and in vitro assessment are shown in Figure 4. Cell proliferation activity assessed using WST-1 assay was similar for rat (Figure 4A) and pig (Figure 4C) liver cells cultured for a short period (Figure 4). The cell morphology (Figure 4B,D) revealed spherical hepatocytes with many cells forming a polygonal shape due to contact with neighboring hepatocytes. Diagonal shape changing and re-establishment of cell–cell contact were observed clearly throughout the culture period. Pig hepatocytes isolated using the Cell Saver^®^ strategy were further confirmed to be functional as they expressed albumin and hepatocyte-specific enzyme cytochrome P450 3A4 (CYP3A4) (Figure 4E). Further semi-quantitative analysis of 6 fields revealed an average of 74.2% cells (total 1125 cells) expressed albumin, and 75.5% (total 1917 cells) expressed CYP3A4. The same was observed in isolated porcine liver cells using the traditional method (Appendix A).

### 3.3. Engraftment Capability after Cell Transplantation through In Vivo Functional Assessment

R+D DPPIV-knockout animal model is a stable testing standard for functional hepatocyte transplantation of acute liver injury in our laboratory. In liver, DPPIV is a cell surface ectopeptidase confined to the bile canalicular domain of the hepatocyte cell surface and to the brush border of interlobular bile ducts and the common bile duct [12]. Long-term repopulation after transplantation supports the production of quality and metabolically active cells [13] using either isolation approach. To follow the principle of Replace, Reduce, and Refine (3Rs) [14] in animal testing and to serve as certainty for quality assurance to test the feasibility using Cell Saver^®^-implemented procedure, hepatocyte transplantation with donor cells isolated from the traditional method employed 2 recipients sacrificed in each time points (at 1 and 2 weeks).

The experimental design for hepatocyte transplantation using an R+D injured DPPIV (−/−) rat model is shown in Appendix A. After 1 week of transplantation, cells with DPPIV expression showed cell colony expansion at 2 weeks and repopulation at 4 weeks (Figure 5A). Engraftment efficiency at weeks 1 and 2 with the Cell Saver^®^ strategy seemed non-significantly lower than that with the traditional method (1 week, *p* = 0.534; 2 weeks, *p* = 0.211) (Figure 5B). The engraftment efficiency of donor hepatocytes isolated from the Cell Saver^®^ strategy at 4 weeks was significantly higher than that at 1 (*p* = 0.016) or 2 (*p* = 0.037) weeks. An active interaction between engrafted donor hepatocytes and native proliferative ductular cells (marked with CK-19) is presented in Figure 5C (arrow and arrowhead). The rescue rates after hepatocyte transplantation were 100% in both groups since D+R model of acute liver injury is not lethal. Gradual recovery of liver histology and inflammation is shown in Figure 6. Global liver functional recovery after hepatocyte transplantation evidenced through serological changes is shown in Figure 7. Rats with hepatocyte transplantation, either with the Cell Saver^®^ strategy or traditional method, showed trends of reduced liver injury markers (aspartate aminotransferase, alanine aminotransferase, lactate dehydrogenase, and total bilirubin) compared with rats with sham operation (Figure 7). The two groups had similar recovery patterns of serological profiles in terms of aforementioned markers, ammonia, urea nitrogen, and international normalized ratio of prothrombin time (Figure 7), with respective *p* values (1 week and 2 week) of (0.406, 0.279), (0.123, 0.658), (0.485, 0.098), (0.316, 0.374), (0.454, 0.931), (0.310, 0.478), and (0.837, 0.677).

## 4. Discussion

This study had three key results. First, with the Cell Saver^®^ strategy, isolated hepatocytes had preserved viability and were the major cell component in the collected cell suspension. Second, these cells, forming cell-cell contact under culture, expressed albumin and CYP3A4 in vitro, functional characteristics of hepatocytes [15,16]. Third, cell therapy for acute liver injured DPPIV knockout rats using hepatocytes isolated using the Cell Saver^®^ strategy provided comparable engraftment and functional recovery to the traditional method. Besides, the whole sterile closed-system processing set from aspiration of liver soup to bag product can be replaced easily if the black filter ever got clogged by tissue debris when much large amount of cells (billions) are processed for human livers. Although we did not perform liver cell isolation of the whole porcine liver by Cell Saver strategy in this pilot study, we believe the convenience and labor-saving nature is ideal for large-scale production.

Automatic liver cell processing to facilitate large-scale clinical application had long been discussed in a preclinical study [17]. Moreover, in clinical cell therapy, minimal manipulation is crucial during the manufacturing process of cell products [18,19,20]. Our study provides a proof-of-concept solution to prevent frequent back-and-forth transfer between the hood and centrifuge in a GMP-approved laboratory, which is lengthy in large-scale processing and had increased contamination risk. Furthermore, if the first part of the cell processing (collagenase perfusion) procedure could be performed in situ clinically, after aspiration into the Cell Saver^®^ system, the subsequent procedures are in a close circuit until the end. This may greatly facilitate cell therapy, particularly hepatocyte transplantation, in liver disease treatment.

In our study, most isolated liver cells obtained using the Cell Saver^®^ strategy were hepatocytes, which were mature parenchymal cells that underwent considerable destruction during acute liver failure. Although proliferative ductular cells emerge to regain liver function [21], the number and functional differentiation of cells are often insufficient. Hepatocyte transplantation is a timely lifesaver. Furthermore, sinusoidal endothelial cells, macrophages, stellate cells, and cholangiocytes, in either simple naked or complex organoid form, have been investigated as other cell sources to transplant for treating liver diseases [22,23,24,25,26,27,28,29]. From the developmental perspective, the construction of liver bud organoids requires “combo” cells from different lineages to reconstitute hepatic, stromal, and endothelial interactions and to start a colony [30,31]. Therefore, although the purification procedures in this study lack Percoll density gradient centrifugation, producing ultrapure hepatocytes is not necessary. In fact, long-term functional hepatocyte engraftment can be observed in an in vivo transplant model.

The limitation of our pilot study is that primary assays of non-human hepatocyte function were performed and in vivo cell transplantation of pig hepatocytes was not tested. The contamination rate in Cell Saver^®^ strategy was not evaluated in this preclinical feasibility study. Moreover, human hepatocyte isolation is significantly more difficult to perform than rodents or porcine liver digestion. Full panel assays of mature liver genes and hepatocyte function including drug-metabolizing enzymes, bile acid synthesis, glycogen storage, urea cycle, serum protein synthesis, cholesterol metabolism, and lipid uptake should be performed in clinical trial assessing human hepatocytes. The proposed technology needs to be validated in a clinical setting using human tissues in the future.

## 5. Conclusions

Although liver cell processing using the Cell Saver^®^ strategy was successful in small number of rounds, the definite clinical feasibility needs to be validated in future large-number studies. Transplanted hepatocytes using this collection strategy had long-term functional engraftment. The accumulated data based on this reduced manipulation strategy in our preclinical study pave the way for future clinical applications.

## Figures and Tables

**Figure 1 jpm-11-00326-f001:**
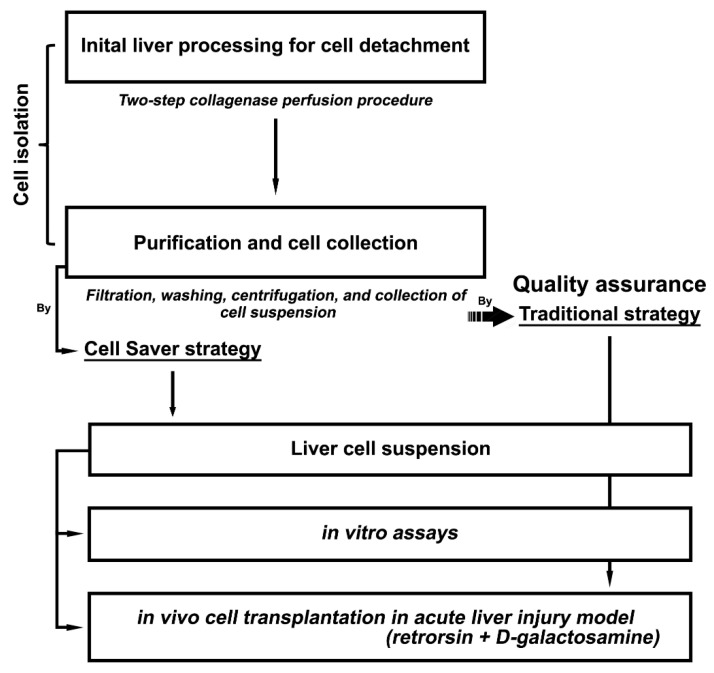
Research and experimental design.

**Figure 2 jpm-11-00326-f002:**
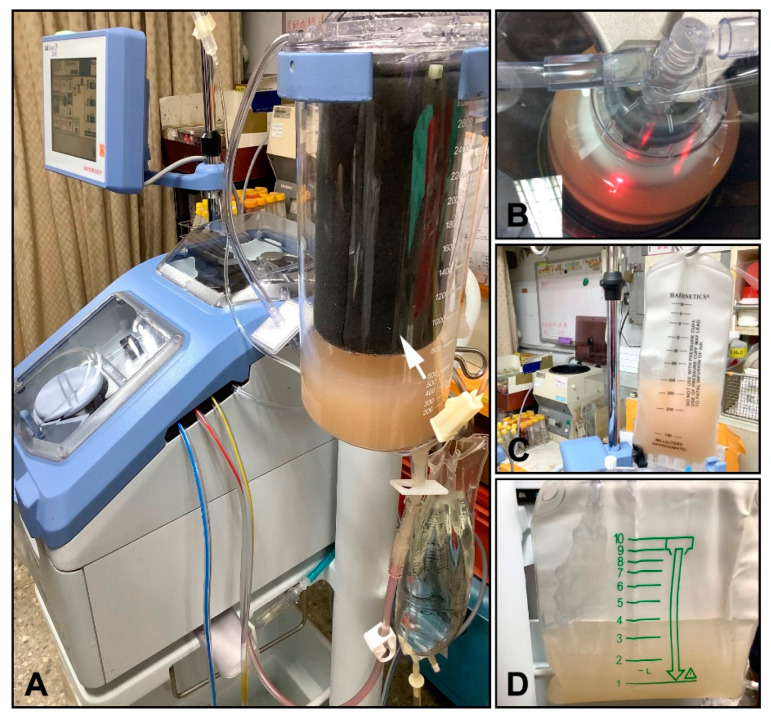
Liver cell isolation through the Cell Saver^®^ strategy. (**A**) Liver cells were filtered through a black sponge (arrow) with an average pore size of 100 μm. (**B**) Cell washing through centrifugation. (**C**) Cell suspension was collected after centrifuged washing. (**D**) Waste washing solution.

**Figure 3 jpm-11-00326-f003:**
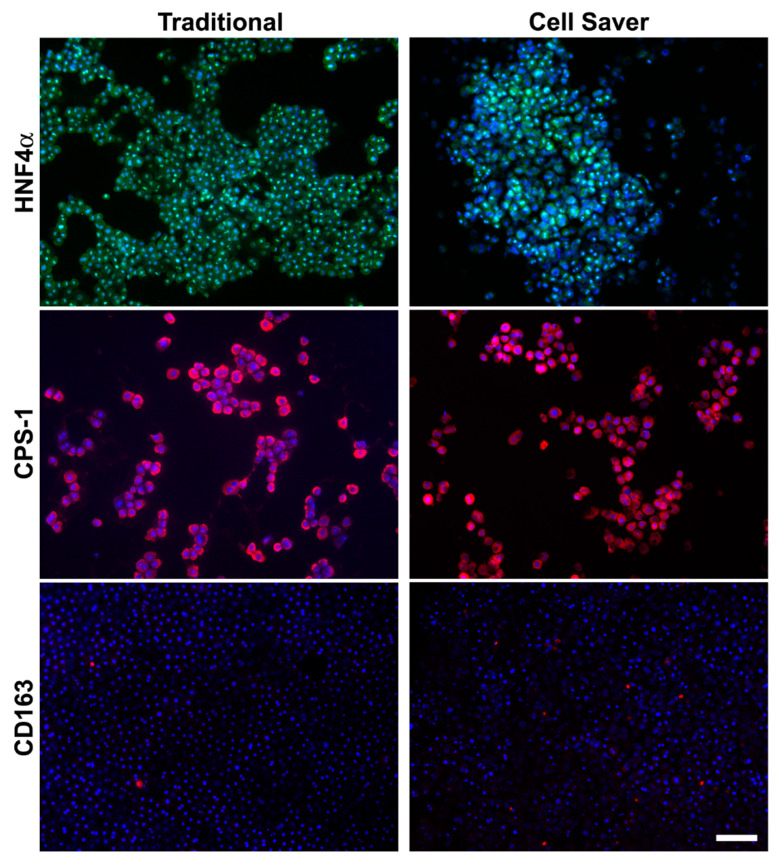
Phenotypic characterization of released cellular products immediately after isolation of rat livers. Majority (>85%) liver cells were mature and functional hepatocytes (positive for hepatocyte nuclear factor 4 alpha (HNF4α) and carbamoyl-phosphate synthase 1 (CPS-1). Few CD163^+^ macrophages (red dots) were seen among isolated cells by either the traditional method and Cell Saver^®^ strategy. Scale bar: 100 μm.

**Figure 4 jpm-11-00326-f004:**
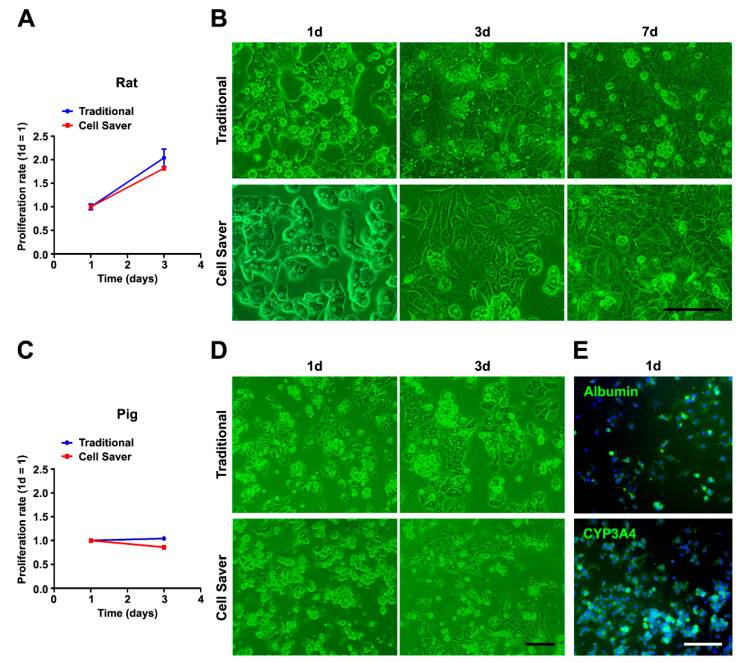
Primary hepatocyte culture, characterization, and cell proliferation assay (**A**,**B**, rat liver cells; **C**–**E**, pig liver cells). (**B**,**D**) Morphology of hepatocyte colony was observed using either the traditional method or Cell Saver^®^ strategy. (**A**,**C**) Cell proliferation of isolated liver cells measured through water soluble tetrazolium-1 (WST-1) colorimetric assay was similar between the two strategies. (**E**) Isolated hepatocytes obtained using the Cell Saver^®^ strategy were confirmed based on albumin and cytochrome P450 3A4 (CYP3A4) expression. Scale bar: 100 μm.

**Figure 5 jpm-11-00326-f005:**
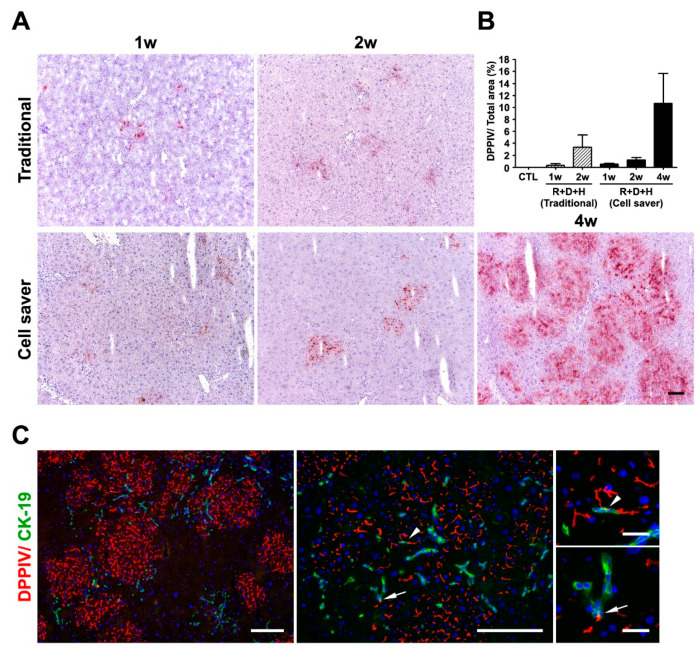
Engraftment and repopulation after hepatocyte transplantation. (**A**) DPPIV histochemical staining for donor hepatocytes showed that donor hepatocytes collected using the Cell Saver^®^ strategy can engraft and repopulate recipient liver well in 4 weeks. (**B**) Engraftment efficiency comparison between the traditional method and Cell Saver^®^ strategy. (**C**) Repopulation of donor hepatocytes derived using the Cell Saver^®^ strategy 4 weeks after transplantation, identified using immunofluorescent staining of DPPIV. Small bile ductular cells were marked using CK-19. Close contact connection of donor hepatocytes and ductular proliferative cells are observed (arrow and arrowhead). R+D+H: retrorsine and D-galactosamine + hepatocyte transplantation. Scale bar: 100 μm.

**Figure 6 jpm-11-00326-f006:**
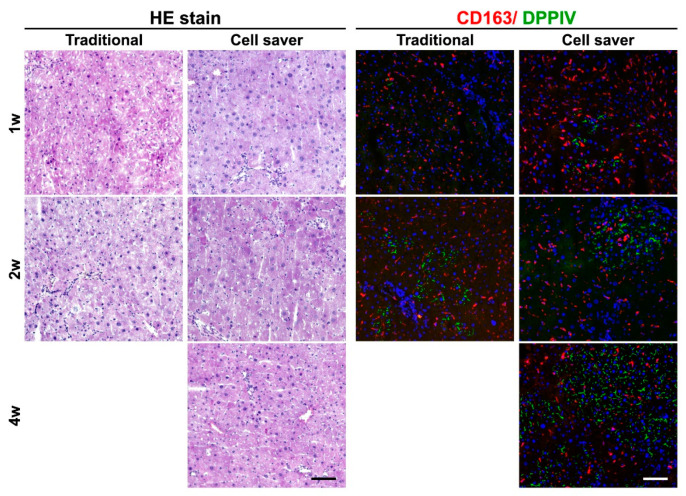
Recovery of liver histology and inflammation after hepatocyte transplantation for acute liver injury. Histological integrity and evenly-distributed macrophage (CD163^+^) could be observed at 2 weeks in both groups. Scale bar: 100 μm.

**Figure 7 jpm-11-00326-f007:**
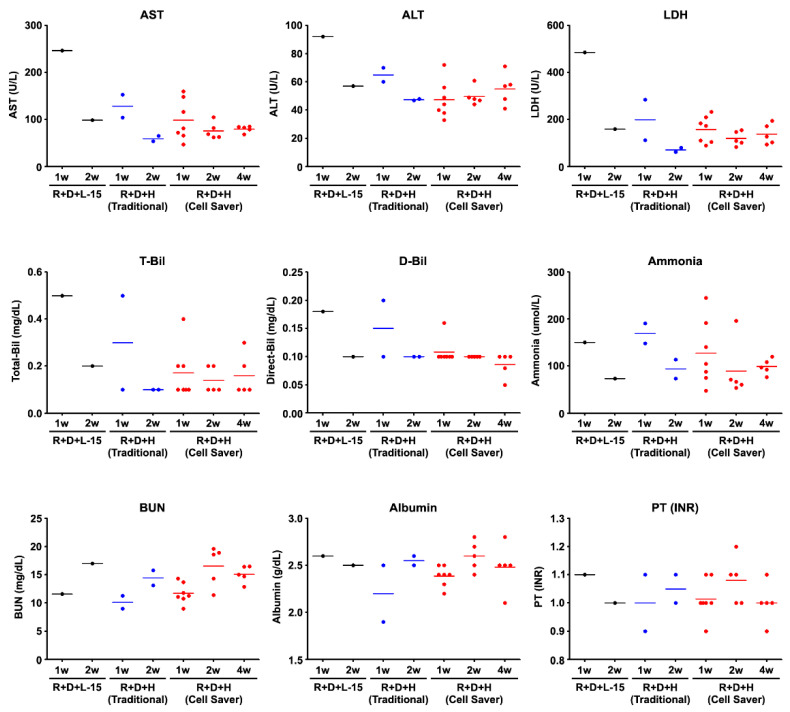
Serological marker changes in recipient rats after hepatocyte transplantation. R, retrorsine; D, D-galactosamine; H, hepatocyte; L-15 as a vehicle control to hepatocyte; AST, aspartate aminotransferase; ALT, alanine aminotransferase; Bil, bilirubin (T, total and D, direct); BUN, blood urea nitrogen; PT(INR), international normalized ratio of prothrombin time; LDH, lactate dehydrogenase.

**Table 1 jpm-11-00326-t001:** Primary antibodies.

Antibodies	Company/Producer	Cat. Number	Dilution
Albumin	Bethyl Laboratories	A90-234A	1:100
CD163	Serotec	MCA342R	1:100
CK-19	Novacastra, Newcastle	NCL-CK19	1:100
CPS-1	Santa Cruz	SC-10516	1:100
CYP3A4	Santa Cruz	SC-27639	1:200
DPPIV	R&D Systems	AF954	1:100
HNF4α	Cell Signaling	#3113	1:100
SE-1	Immuno-Biological Laboratories	10078	1:100

## Data Availability

All data generated or analyzed during the current study are included in this published article.

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
