# Peer review of "Preclinical Application of Reduced Manipulated Processing Strategy to Collect Transplantable Hepatocytes: A Pilot and Feasibility Study"

_jpm, 2021, doi:10.3390/jpm11050326_

Round 1
Reviewer 1 Report
Chen and co-authors describe a proof-of-concept preclinical study, where primary hepatocyte have been isolated with the support of an intraoperative cell recovery machine (commonly used for autologous red blood cell transfusion) and analysed in vitro and infused in model for acute liver injury.
Revision and improvement in hepatocyte isolation procedure have been attempted and offered for the past 50 years. Critical and detailed analysis between “classical” and new procedure have been compared to prove superior effects and improved outcome. The authors performed isolation of non-human hepatocytes only, and it is unclear how many different isolations and if any of these have been correctly performed side-by-side, using “classical” isolation technique or revised Cell Saver-implemented procedure. Liver cells isolated by traditional method have been apparently obtained by 2 donors only, while CellSaver isolations are frequently represented as 7 dots (Figure 5) and sometimes less. Please motivate
The current manuscript aims to offer revised and facilitated cell isolation procedure Cell Saver Elite device. The study may constitute an interesting and potential technical improvement to current the current clinical-grade isolation of liver cells, if successfully proven. Importantly, the proposed technology needs to be validated in a clinical settings using human tissues. Human hepatocyte isolation is significantly more difficult to perform than rodents or porcine liver digestion.
Moving from the study summary offered in the first line of Discussion section (page 8), the major concerns for the proposed study are:
- The authors stated ”with the Cell Saver strategy, isolated hepatocytes had preserved viability and were the major cell component in the collected cell suspension”: Such conclusion should be supported by real measurement (viability, recovery and cell product phenotype characterization by immunocytostaining techniques)
- Cell Saver-derived cells ”forming cell–cell contact under culture, expressed albumin and CYP3A4 in vitro”. Albumin and CYP3A4 staining are only indicative of quality and maturaty level of isolated cells. Once again, side-by-side comparison between classical technique and CellSaver technology should be reported.
- “cell therapy for acute liver injured rats using hepatocytes isolated using the Cell Saver strategy provided comparable engraftment and functional recovery to the traditional method”. Is this another way to support the production of quality and metabolically active cells using both isolation approaches?
- CellSaver “minimally manipulated processing strategy is ideal for scaling-up to large animals”. This final statement is quite unspecific and requires additional discussion. First of all, why CellSaver-based isolation has been described as minimally manipulated processing? Second, why the authors suggest that implementing the hepatocyte isolation technique with CellSaver washing device would support large-scale production is not clear. Please elaborate
At lines 194-195, the authors describe as “The engraftment efficiency of donor hepatocytes isolated from the Cell Saver strategy at 4 weeks was significantly higher than that at 1 (P = 0.016) or 2 (P = 0.037) weeks”. The donor cell engraftment and integration within parenchyma has been shown to occur within 24-48 hrs after injection. Any cells left in the circulatory system are rapidly removed by macrophages. And different preclinical approaches have been attempted to improve hepatocyte engraftment (i.e., vasodilators, or specific inhibitors of macrophage function). Thus, what the author measured 7 or 14 days after injection was actually the level of survival or cell duplication. However, direct comparison between CellSaver-isolated hepatocyte and traditional technology products is required and the cell products should be compared and analysed.
Results mentioned at page 6 (lines 200-205) need to be supported by graphs and real data.
Another technical concern for future translation to clinical practice is in relation with the filtration process: human livers result in large amount of cells (billions), and they need necessarily to be filtered; can the black filter depicted in Figure 2 be eventually replaced during the process?
The authors interestingly sustain that Cell Saver strategy results in isolated hepatocytes with preserved viability, and characterized by hepatocyte characteristics. Side-by-side analysis on cell viability, recovery and phenotypic characterization for cellular products resulting from “classical” technique in comparison with Cell Saver strategy need to be performed. FACS or immunocytological analysis immediately after isolation need to be implemented.
Unfortunately, the current manuscript limits observation to immunocytological staining for albumin and CYP3A4 in vitro. Functional analysis for the released products, preferably using in vitro assays for different hepatic functions (phase 1 to 3, and urea metabolism) should validate the final cell suspension before infusion. Engraftment and functional activity in isolated cells pre- and post-transplant are highly recommended.
Finally, acute liver injured rats rescued by isolated hepatocytes collected using “classical” and CellSaver-implemented procedure need to be performed and accurately compared. Rescue rate as well as long-term survival (and related histological analysis) and inflammation should be included.
Reviewer 2 Report
Overall comments to the Author
Thank you for the opportunity to review the manuscript entitled, "Preclinical application of minimally manipulated processing strategy to collect transplantable hepatocytes: a pilot and feasibility study". The concept of this study is interesting and clinically important, and the manuscript is well written. However, this study contains several drawbacks to be mentioned. I have the following comments for major revision:
Major
- We know that the viability of isolated hepatocytes is difficult to be maintained. If the quality of the isolated hepatocytes using Cell Saver are satisfactory, this strategy can be a standard isolation technique. Therefore, the assessment of cell viability is essential in this study. The authors should describe more about the methods and results regarding the analyses of cell viability. How many livers from rats and pigs were isolated and analyzed? In the results, the authors described that hepatocytes constituted the major cell population (>85%) and the viability was >80%. How about the number of hepatocytes obtained by Cell Saver? The details of these results should be clearly indicated.
Minor
- How about the contamination rate in Cell Saver strategy? Is it reduced by the strategy?
Reviewer 3 Report
REVIEW: "Preclinical application of minimally manipulated processing strategy to collect transplantable hepatocytes: a pilot and feasibility study".
The authors of the current article describe method of possible transplantion that can be useful in liver tranplantation. The authors use rat and pig liver cells and then cultured 18 in vitro, and their hepatocyte-specific enzymes for examination. After that process, they transplanted the 19 hepatocytes in an established acute liver–injured (retrorsine+D-galactosamine-treated) rat model for 20 engraftment. The authors examined then liver tissue 1, 2, and 4 weeks after transplantation followed by do- 21 nor-cell identification and histological, serologic, and immunohistopathological examination. The authors examined in vitro and in vivo 16 hepatocyte isolation feasibility and cell function of Cell Saver Elite.
The authors concluded their work that Cell Saver requires minimal manipulation for isolating transplantable hepatocytes.
All the paragraphs of the article has been described correctly and clear.
Author Response
Thank you for providing positive feedback.
Round 2
Reviewer 1 Report
The (human) hepatocyte isolation procedure has required decades of work, preclinically and later clinical, to reach level of quality both in term of yield and cell viability / function. The heterogeneity of human liver tissue cannot be controlled, but the conditions and reagents for hepatocyte isolation can and need to be standardized, and yet, continually improved by careful attention to each step of the isolation process. Still margins of improvement are available, but accurate comparison is needed to improve such critical and important (clinical) procedure.
R1 version is improved but still requires critical revision. Based on the authors declare simple intention to “test the feasibility of Cell Save strategy, with classical strategy as standards”, I would strongly recommend modifying the title, at first. The current title (“Preclinical application of minimally manipulated processing strategy to collect transplantable hepatocytes: a pilot and feasibility study”) is misleading and incorrect. Minimal manipulation notion is conceptually different and absolutely not pertinent in the current context. The authors affirmed as “in most jurisdictions, minimal manipulation is defined as acts that do not result in fundamental changes to the structure or biological characteristics of the original cells or tissues”. Such critical difference in take-home message needs to be better clarify and elaborated. Nevertheless, the selection of words is critical and need to be consistent with GMP terminology. Maybe the authors should rather refer to “common laboratory equipment” or procedure that do not result in permanent changes in structure or functional characteristics of the isolated cells
The authors clearly stated (in reviewers’ point-by-point replies, but less clearly in the submitted manuscript) that their intent is to report a pilot study, where feasibility and adjustment of current mechano-enzymatic procedure around a ”common” lab equipment (Cell Saver) rather than offering an accurate and properly tested revised procedures to isolate primary hepatocyte. Consequently, the authors should clearly state from the very beginning (Introduction and Abstract) that they do not intended to compare Cell Save-based strategy with traditional procedure.
Furthermore, the number of hepatocyte isolation procedures reported (using Cell Saver device) have been performed on one swine donor only (twice, exploiting the split liver technique most likely) and on 3 rats (on whole liver). Transplantation of the cell products in model for ALF was limitedly performed using rodent cells (n=2). Authors’ justification based on 3R principle is questionable. However, if the authors include and comment such decision in the manuscript, I think it’s acceptable (reader should be fully aware of such strategy). Justification and explanation for every authors’ decision and experimental approach should be included in the manuscript
Interestingly, (rat) hepatocytes isolated using Cell Savor procedure was not different nor superior in viability/quality compared to classical procedure. Unfortunately, swine hepatocyte isolation has not been offered (despite the split liver procedure would be extremely instrumental and useful to properly compare classical vs revised procedure). Please comment and elaborate
The authors found sufficient stain hepatocyte for Albumin and CYP3A4 enzyme to determine quality and maturity level of isolated cells. Such minimal criteria to release cellular protocols has been previously described in old studies, characterized by low level of competence and quality. Although the quality of the hepatocytes used for cell transplantation is critical, no consensus currently exists on protocols to assess the function of hepatocytes prior to infusion. Traditional assays for measuring metabolic functions in primary hepatocytes (i.e., testosterone metabolism) frequently involve highly technical equipment, time-consuming methods and large numbers of cells. During the past years, several groups described and optimized in vitro methods aimed to quickly assess metabolic capabilities of primary hepatocytes, frequently coupled with long-term function, after exposure to prototypical inducing agents (omeprazole, phenobarbital and rifampicin).
Such assays rapidly and easily measure Cytochrome P450 activities, phase 2 conjugation, ammonia metabolism in addition to ”classical” viability/apoptosis and cell yield. AFP and CYP3A7 enzyme are commonly considered fetal markers, while Albumin, CYP450 enzymes and transporters are critical for the metabolism of endogenous and exogenous compounds in adult and functional primary liver cells. Commonly used markers for liver cell characterization are CYP3A4 and CYP3A7 which are strongly age-dependent. CYP3A7 is an enzyme required for the fetal/neonatal growth and development, and the excretion of endogenous compounds. Its amounts decrease during the first postnatal years to low adult levels with a concomitant increase in CYP3A7 expression that may mature levels in the adolescence. Data on ALB and AFP expression, which are mature and fetal markers respectively, indicated towards the same direction. Moreover, transcription and growth factors that are known to be crucial for the regulation of mature liver genes (i.e. HNF1, HNF3, HNF4, HNF6, LXRa, FXR) may also be relevant for precise and punctual characterization of the final product. Above all, drug-metabolizing enzymes are essential (CYP1A1, CYP1A2, CYP2A6, CYP2B6, CYP2C8, CYP2C9, CYP2C19, CYP2D6, CYP3A4, CYP3A7), with CYP3A4 being the most abundant in adult liver, while in general those belonging to families 1-3 are responsible for 70-80% of phase-1 metabolism (Chen, C., et al. Biotechnology Challenges to In Vitro Maturation of Hepatic Stem Cells. Gastroenterology 154, 1258-1272, doi:10.1053/j.gastro.2018.01.066 (2018)). These can be evaluated based on gene expression levels or preferably tested on intact cell product in vitro (Gramignoli, R. et al. Rapid and sensitive assessment of human hepatocyte functions. Cell Transplant 23, 1545-1556, doi:10.3727/096368914X680064 (2014); Gomez-Lechon, M. J.; Castell, J. V.; Donato, M. T. Hepatocytes--the choice to investigate drug metabolism and toxicity in man: in vitro variability as a reflection of in vivo. Chem. Biol. Interact. 168(1):30-50 (2007)). Other in vitro assessments that could serve as indicators of hepatic maturity are bile acid synthesis (formation of cholic acid and chenodeoxycholic acid), glycogen storage (periodic acid-Schiff staining for glycogen), urea cycle (ammonia metabolism or/and urea production), as well as serum protein synthesis such as alpha-1-antittrypsin, fibronectin, transferrin, coagulation factors and others. Moreover, criteria of differentiation efficiency are also considered cholesterol metabolism (formation of 7 α-hydroxycholesterol) and lipid uptake (transport of low-density lipoprotein).
The authors limit their evaluation to albumin and CYP3A4 stainings, unfortunately,with low magnification and limited analysis (how many fields/cells have been counted?). Plenty of cells and clusters will severely limit discrimination and evaluation
Finally, hepatocytes have also been stained for HNF-4α and CPS-1. Macrophages were marked with CD163 expression. Additional markers for cholangiocyte and endothelial / stellate cells would be highly relevant to qualify final product.
Reviewer 2 Report
Overall comments to the Author
Thank you for the opportunity to review the revised manuscript entitled, "Preclinical application of minimally manipulated processing strategy to collect transplantable hepatocytes: a pilot and feasibility study". The authors fully revised manuscript and correctly answered my questions. However, the number of isolation procedures should be increased because only 2 for pig donors and 3 for rat donors are not enough to assess the feasibility of this promising strategy.
Author Response
Dear Reviewer 2,
We thank you once again for reminding us the conclusion about the feasibility of this promising strategy. We had revised the conclusion remarks as "Although liver cell processing using the Cell Saver strategy is successful in small number of rounds, the definite feasibility needs to be validated in future large-number studies." in Page 11, line 314-315. Thank you.
Sincerely,
Cheng-Maw Ho